# Hydrophilic Films Based on Carboxymethylated Derivatives of Starch and Cellulose

**DOI:** 10.3390/polym12112447

**Published:** 2020-10-22

**Authors:** Katarzyna Wilpiszewska, Adrian Krzysztof Antosik, Beata Schmidt, Jolanta Janik, Joanna Rokicka

**Affiliations:** Department of Chemical Organic Technology and Polymeric Materials, Faculty of Chemical Technology and Engineering, West Pomeranian University of Technology Szczecin, 70-322 Szczecin, Poland; lunatyk88@interia.pl (A.K.A.); beata.schmidt@zut.edu.pl (B.S.); jolanta.janik@zut.edu.pl (J.J.); joanna.rokicka@zut.edu.pl (J.R.)

**Keywords:** carboxymethyl starch, carboxymethyl cellulose, hydrophilic films, polysaccharide films

## Abstract

The carboxymethylated derivatives of starch (CMS) and cellulose (CMC) were used for film preparation. The infrared spectroscopy revealed that crosslinking via ester bridges with citric acid occurred between the two polysaccharide derivatives. The effect of polysaccharide derivatives ratio on physicochemical properties of prepared films was evaluated. Generally, the values of tested parameters (moisture absorption, surface roughness, and mechanical and thermal properties) were between the values noted for neat CMS or CMC-based films. However, the physicochemical properties of the system with equal CMS/CMC weight ratio diverged from this trend, i.e., the highest tensile strength, the highest Young’s modulus (ca. 3.4 MPa and ca. 4.9 MPa, respectively), with simultaneously the lowest moisture absorption (18.5% after 72 h) have been noted. Such systems could potentially find application in agriculture or pharmacy.

## 1. Introduction

The growing interest in the polysaccharide-based films is observed in recent decade. Starch and cellulose are the most abundant, non-toxic, and biodegradable biopolymers. Their carboxymethylation (by substitution of hydroxyl groups with monochloroacetic acid or sodium monochloroacetate, in the presence of strong alkali) results in obtaining ionic ether derivatives: carboxymethyl starch (CMS) and carboxymethyl cellulose (CMC), respectively. Unlike native starch and cellulose, these derivatives are soluble in cold water, however, their physicochemical properties depend greatly on degree of substitution (DS), which is the average number of hydroxyl groups substituted in a recurrent polysaccharide unit [1]. The carboxymethylated polysaccharide derivatives are used in many fields of application, like food, cosmetics, pharmaceutical, or paper industries [2].

Carboxymethyl starch has been reported as a hydrophilic film forming polymer [3]. Non-crosslinked CMS-based films are soluble in cold water [4,5]. Applying crosslinking agent, such as: sodium trimetaphosphate [6], sodium hexametaphosphate [7], dichloroacetic acid [8], or multifunctional carboxylic acids [9,10] allowed to obtained hydrogel material with various swelling capacity. The mechanical strength of CMS-based films could be improved by introducing clay nanoparticles [7,10] or by blending CMS with other polymer, e.g., starch [11] or zein [12]. Blending CMS with CMC would allow to impart new functional properties, e.g., biodegradable material with modified hydrophilicity, which could be beneficial for agricultural purposes or desirable for active food packaging [13], edible coating [14,15], as well as in pharmacy (drug carrier) [16,17]. As both derivatives are hydrophilic, their thermodynamic miscibility could be expected. Moreover, CMC is the ether derivative of high molecular weight fibrous cellulose, thus its presence could result in mechanical properties improvement [18]. Many reports refer to preparing CMC blends, very often with starch [19,20,21,22,23]. The addition of CMC into rice starch-based film plasticized with glycerol resulted in improvement of mechanical properties, transparency as well as thermal stability [19]. The simultaneous extrusion of corn starch with 5% w/w CMC resulted in instant gels with higher firmness and stability when compared to starch material without cellulose derivative [22]. Introducing CMC into glycerol-plasticized sorghum starch allowed to prepare films with enhanced water resistance [24]. Similarly, Tavares et al. [20] reported improved moisture barrier properties as well as increased maximum tensile strength of both corn and cassava starch-based systems containing 50% w/w CMC.

Although CMS/CMC system has been recently proposed for pressure sensitive adhesives applications [25,26] the systematic study on preparing and determining the properties of crosslinked CMS/CMC films has not been reported so far.

In this paper, the films based on CMS and CMC (using glycerol and citric acid as plasticizer and crosslinking agent, respectively) have been prepared. Citric acid (CA) is an organic acid widely existing in citrus fruits, containing three carboxylic groups and one hydroxyl that could interact with OH and carboxymethyl groups of polysaccharide derivatives. For film preparation high substituted CMS (DS 0.8) and CMC (2.6) have been applied. The effect of CMS/CMC ratio on the physicochemical properties (moisture absorption, morphology, as well as thermal and mechanical properties) of prepared films has been determined.

The obtained CMS/CMC films are edible, however their potential application in agriculture or pharmacy (as a multilayer dressing materials or in transdermal systems) is planned.

## 2. Materials and Methods

### 2.1. Materials

Potato starch (13.6–14 wt% moisture) was purchased from Nowamyl S.A. (Nowogard, Poland). Monochloroacetic acid (MCA, 98.5%, Chempur, Piekary Śląskie, Poland) was used as an etherifying agent, whereas isopropanol (99%, Chempur, Piekary Śląskie, Poland) was used as a reaction medium. Glycerol (98%), citric acid monohydrate (CA, 99%), sodium hydroxide (microgranules, 98%), acetic acid, and copper sulfate penthahydrate were the products of Chempur (Piekary Śląskie, Poland), murexide and ethylenediaminetetraacetic acid disodium salt dehydrate (EDTA) of Sigma-Aldrich (Taufkirchen, Germany). Carboxymethyl cellulose (DS 2.6) was the product of Pronicel Sp. z o.o. (Warszawa, Poland).

### 2.2. Preparation of CMS with High Degree of Substitution

Modification of potato starch was carried out in a batch reactor equipped with a mechanical stirrer, a thermocouple, and a capillary tube supplying nitrogen to the reaction system. Starch was etherified in isopropanol/water in a one-step process. In the batch reactor, MCA (35 g) was dissolved in isopropanol, and then aqueous solution of NaOH was added (the molar ratio of MCA/polysaccharide recurrent unit was 2, whereas NaOH/MCA 2.2). When the mixture became white and homogeneous, starch (30 g) and remaining NaOH were introduced. Reaction was performed for 2.5 h at 50 °C. Obtained product was filtered, neutralized with glacial acetic acid, washed five times in 80 wt% methanol aqueous solution, and then washed once again in methanol and dried in the air.

Degree of substitution was determined according to the method described in other work [27]. The CMS sample was moisturized by 1 mL of ethanol and dissolved in 50 mL of distilled water. Subsequently, buffer was added (NH_4_Cl 0.187 M aqueous solution, 20 mL), neutral pH was adjusted, and then the whole mixture was poured into a measuring flask (250 mL) with 50 mL CuSO_4_ (0.039 M) solution. After 15 min, the measuring flask was filled up with water and the whole content was filtered. Filtrate was titrated with 0.05 M EDTA solution using murexide as an indicator.

### 2.3. Preparation of CMS/CMC Films

Into 100 mL of distilled water 3 g of CMS/CMC dry powder mixture (with CMS to CMC ratio: 0/100, 20/80, 40/60, 50/50, 60/40, 80/20, or 100/0 wt%) was dissolved. The molar ratios for CMS/CMC: 20/80, 40/60, 50/50, 60/40, 80/20 systems were: 0.36, 0.98, 1.48, 2.34, and 5.96, respectively. Subsequently, 2 g glycerol (plasticizer) and 2 g citric acid (crosslinking agent) was added and stirred until homogeneity. Then, the solution was mixed gently (to remove air bubbles) and poured into PTFE mold (15 × 15 cm^2^) and dried for 48 h at 60 °C. Obtained film (thickness 200–300 μm) was peeled off and used for further tests.

### 2.4. Methods

The FTIR analyses of the films were performed in Nexus FTIR Spectrometer Thermo Nicolet (Waltham, MA, USA) with Golden Gate ATR attachment. The resulting spectra were converted using the software OMNIC. Before measurement, the film was immersed in distilled water for 24 h at room temperature to remove residual citric acid and dried at 50 °C [28].

The laser scanning microscopy (LSM) measurements were conducted with application of VK-9700 microscope (Keyence, Mechelen, Belgium). The microscope was equipped with a short wavelength (408 nm) laser light source and a pinhole confocal optical system, with 400× magnification. During LSM analysis the field of the microscope was scanned using a laser beam and an X–Y scan optical system. A light reflected from each pixel in the field of view was detected by the light receiving element. While moving the objective lens in the *Z*-axis and repeatedly scanning the measured area the reflecting light intensity based on the Z position was obtained.

Moisture absorption tests were performed as followed: for each film three squares (1.5 × 1.5 cm^2^) were cut and dried in desiccator for two weeks. Subsequently, dry samples were weighted and placed into climatic chamber (55 ± 2% humidity, 25 ± 2 °C) (Memmert HCP 135, Buechenbach, Germany), the weight of tested samples was controlled after: 3, 5, 7, 24, 48, and 72 h. Moisture absorption was calculated using equation [18]
(1)$$A_{t}=\frac{M_{t}-M_{0}}{M_{0}}\cdot 100\%$$
where *A_t_*—moisture absorption after time t [%]; *M_0_*—mass of dry sample [g]; *M_t_*—mass of sample after time *t*: 3, 5, 7, 24, 48, and 72 h, respectively [g].

The dynamic mechanical thermal analyses (DMTA) of the CMS/CMC films were determined using DMTA Q800 (TA Instruments, New Castle, DE, USA). The measurements were carried with film tension clamp at frequency of 1 Hz, heating rate 3 °C/min, and temperature range from −70 to 170 °C.

The mechanical properties of the CMS/CMC films were determined using a tensile tester (Instron 4026, Instron Corporation, Marwood, MA, USA) equipped with 1 kN load cell. The specimens (10 × 100 mm^2^ strips) were conditioned at RH = 55% for 24 h in a climatic chamber (Memmert HCP 135, Buechenbach, Germany). The initial grip separation and cross-head speed were 50 mm and 1 mm/min, respectively. The true strain ε was determined by ε = ln(*L*/*L*_0_), where *L* and *L*_0_ (mm) were the length during the test and the length at zero time, respectively. The true stress σ was calculated by σ = *F*/*S*, where *F* (Pa) was the applied load and *S* (mm^2^) the cross-section area. As *S* was determined assuming that the total volume remained constant, so *S* = *S*_0_*L*_0_/*L*, where *S*_0_ was the initial cross-sectional area. The stress–strain curves were plotted and the tensile strength as well as Young’s modulus were determined from the slope of the strain region in the vicinity of σ = ε = 0 ([dσ/dε]_ε__→0_) [29]. The mechanical tensile data were averaged over 10 specimens.

## 3. Results and Discussion

### 3.1. Fourier Transform Infrared Spectroscopy (FTIR)

In Figure 1 the FTIR spectra of citric acid, neat carboxymethyl starch, neat carboxymethyl cellulose, and CMS/CMC-based films were presented (for the sake of clarity only the spectra of the CMS/CMC 40/60, 50/50, and 60/40 systems were shown). The polysaccharide derivatives films were washed to remove residual citric acid before testing [28].

The spectrum of citric acid showed a broad characteristic pattern with maximum absorption intensity at ca. 3300 cm^−1^ attributed to –OH stretching [16] and a sharp C-O band at about 1700 cm^−1^ assigned to protonated carboxylic groups [10]. The bands between 1300 and 1070 cm^−1^ could be ascribed to the oscillations of the C–OH group. The spectra of polysaccharide derivatives showed a wide absorption band between 3600–3000 cm^−1^ and at 2900 cm^−1^ attributed to hydroxyl groups and to CH_2_ stretching vibrations, respectively [1]. The bands at 1440 and 1325 cm^−1^ were assigned to CH_2_ scissoring and OH bending vibrations, respectively. The adsorption band at ca. 1100 cm^−1^ characteristic for anhydroglucose O–C stretching bands [20] could be observed. The shift of this peak toward lower wavenumbers could be assigned to the stability of hydrogen bonds formed between polysaccharide molecules [30]. Thus, as for CMS and CMC the wavenumbers are 1008 cm^−1^ and 1020 cm^−1^, respectively, it could be concluded that between unmodified CMS molecules stronger hydrogen bonds were formed than between CMC ones (Figure 1A,E, respectively). The reason could be much higher degree of substitution of cellulose derivative enhancing the steric repulsion. Interestingly, the wavenumbers of this band for CMS/CMC films were shifted toward lower values, and generally decreased with the CMS content increase (1008–1012 cm^−1^), indicating more stable hydrogen bond formation between CMS and CMC when compared to neat CMC system. The band at ca. 1200 cm^−1^ indicates CA addition, and is assigned to C–OH oscillations, as from steric viewpoint more likely is reaction of one from three COOH groups than one OH (maintains unmodified).

The protonated carboxylic groups in the spectra of neat CMS and CMC give strong absorption band at about 1600 cm^−1^ (intensity of this band strongly relates to the DS value) [31]. In the case of CMS/CMC films, the carbonyl group band shifted to 1720 cm^−1^ indicating chemical linkages between polysaccharide derivatives and citric acid (crosslinking agent) via ester bonds [16]. However, the absorption band at ca. 1600 cm^−1^ was still observed, thus it could be concluded that some carboxylic groups remained unreacted. The crosslinking reaction with citric acid, i.e., formation of intermolecular diesters, was presented in Figure 2 [32,33].

It is known that the ester bond formation is catalyzed by low pH [32]. However, because of steric repulsion, it is unlikely that all carboxyl groups of CA could react with polysaccharide derivatives. Moreover, the crosslinking reaction between CMS molecules, as well as between CMC ones could not be omitted (Figure 3), especially when one of the derivatives is in majority in the film forming system.

It is worth to mentioned that the water solubility tests of prepared CMS/CMC films have been performed. The 1.5 × 1.5 cm^2^ film samples were placed in 50 mL distilled water, at room temperature. After 48 h, all the samples maintained their integrity, i.e., they did not dissolve or break apart during the tests. This indirectly indicates successful crosslinking taking account that both CMS and CMC are soluble even in cold water.

### 3.2. Morphology of CMS/CMC Films

The photograph of the film sample was presented in Figure 4—the prepared films were elastic and transparent.

The morphology of all prepared CMS/CMC films using laser scanning microscopy was presented in Figure 5. Additionally, the 3D topographical views were also presented. Only the film based on CMS exhibited relatively smooth surface, which could be observed clearly in the 3D view. Slightly higher surface roughness could be noticed for the CMC-based system. However, for other films the surface was more developed, and it was difficult to find the correlation with the CMS content. The roughness measurements have been performed (Figure 6). The *R*_z_ values were determined as sum of the height of the highest profile peak and the depth of the deepest profile valley within an individual measuring distance, whereas *R*_a_ was the arithmetical mean value of the amounts of the ordinate value within an individual measuring distance. The highest *R*_z_ and *R*_a_ values were determined for CMS/CMC 50/50 system (ca. 102.1 μm and 12.3 μm, respectively). The lowest values were noted for CMS/CMC: 100/0 and 0/100 systems—*R*_z_: 11.6 μm and 30.1 μm, respectively, *R*_a_: 2.1 μm and 3.7 μm, respectively.

Generally, the systems containing both polysaccharide derivatives exhibited noticeably higher roughness values than the neat systems. It could be explained by the uneven distribution of CMS and CMC molecules in the system, as well as inhomogeneous crosslinking density of both polysaccharide derivatives (crosslinking between CMS-CMS or CMC-CMC was possible as well, especially when one polysaccharide derivative was in excess), caused by high viscosity of CMS and CMC solutions, and the differences in DS value of both carboxymethylated derivatives.

### 3.3. Moisture Absorption

The moisture absorption ability for all prepared films as a function of storage time upon conditioning at 55% RH was presented in Figure 7. The value of this parameter raised intensively for ca. 7 h (ca. 24 h in case of CMS/CMC 0/100 film) and after ca. 24 h tended to the balance. Although both, CMS and CMC exhibit hydrophilic character, the moisture absorption clearly correlated with CMC, i.e., increased with its content (the highest value ca. 23.5% noted after 72 h for the CMS/CMC 0/100 film). For comparison, the systems with predominant CMS content exhibited the moisture absorption ca. 20 ± 0.5% after 72 h. It is known that the hydrophilicity of carboxymethylated polysaccharides increases with the DS value increase. The degree of substitution for CMC and CMS used were: 2.6 and 0.8, respectively, thus for every 100 anhydroglucose rings there are 80 carboxymethyl groups located on the CMS chain, whereas on CMC chain 3 times more, i.e., 260 ones. As a result, CMC forms more spatial structure than CMS. After crosslinking (even partial), the polymer net is formed enabling to entrap much more water than crosslinked CMS-based one [25]. This explains the higher absorption capacity of the system containing CMC >50%. This could also explain why the lower CMC content results in moisture absorption decrease.

Ghanbarzadeh et al. [34] observed other effect of CMC addition into starch/CMC system (DS of CMC not given)—moisture absorption decreased with CMC content. However, when a higher CMC amount (20 wt%) has been added, the value of this parameter slightly increased indicating limited improvement of water resistance for higher cellulose derivative loading.

For the films based on rice starch and CMC derived from durian the rind (DS 0.92), the swelling ratio noted for blends was also higher than for film without CMC [19]. No crosslinking agent was applied. The increase of swelling ratio value was attributed to hydrogen bond connections, resulting in creating a large framework able to store the water.

Surprisingly, the film with equal CMS and CMC content (CMS/CMC 50/50) featured the lowest moisture absorption value, i.e., 18.5% after 72 h. As the value of this parameter is determined by the molecular structure and crosslinking [35], it is very probable that for this special ratio CMS/CMC 50/50 the CMS and CMC chains were arranged in the way that favors the crosslinking reaction more than for other CMS/CMC ratios. As a result, a more coherent polymer structure exhibiting lower moisture absorption capability could be formed. Interestingly, the film basing on the CMS/CMC 50/50 blend (prepared as described in Section 2.3, plasticized, but without crosslinking agent addition) was obtained. The moisture absorption of such a film significantly increased within the first hours, and after 24 h was 56% (data not shown in Figure 7, as the value was off the scale). This comparison directly indicates the crosslinking effect in polysaccharide films.

### 3.4. Dynamic Mechanical Thermal Analysis (DMTA)

The evaluation of the loss factor (tan δ as a function of temperature for CMS/CMC films was shown in Figure 8. The loss factor is sensitive to molecular motion and its peak relates to the glass transition temperature. All the curves revealed transition in the range from ca. 21 up to ca. 31 °C. Moreover, for all tested films the signal at ca. −49 °C was noted. It could be assigned to polysaccharide derivatives-poor phase referred to plasticizer transition [10].

Interestingly, additional transition (at ca. 2 °C) was observed for the systems with high CMC content (>50%) indicating the presence of two phases (each phase exhibited its own glass-rubber transition), which could suggest limited miscibility of both polysaccharide derivatives [34]. However, it was noted also for the CMC-based film what means that inhomogeneous crosslinking of carboxymethylated cellulose molecules occurred resulting in phases with various mobility of the molecules, i.e., various crosslinking densities. Importantly, the films with high CMS content did not exhibit this phenomenon. Additionally, their *T*_g_ values were higher than for CMC-reach systems suggesting limited mobility of starch derivative chains that could be affected by crosslinking with CA. Generally, the glass transition values decreased with CMC content increase. However, the reports on starch/CMC systems revealed contrary effect of CMC presence on the *T*_g_ value (determined by DSC method), i.e., *T*_g_ increased with CMC content increase from 5 wt% up to 20 wt% [34].

It should be pointed out that again CMS/CMC 50/50 wt% system was in contrast to the general trend with *T*_g_ ca. 28 °C, i.e., higher than these noted for CMS/CMC: 20/80 and 80/20 systems, respectively; however, inhomogeneous crosslinking could be observed. These results are in correlation with the results of moisture absorption.

### 3.5. Mechanical Properties

The mechanical properties of CMS/CMC films were characterized by tensile measurements at room temperature. The typical true stress–true strain curves were shown in Figure 9. The true stress regularly increased with the true strain up to film break. No necking phenomenon could be observed in the curve indicating relatively good compatibility between biopolymers. The dependence of CMS/CMC ratio on tensile strength, Young’s modulus, as well as elongation at break was presented in Figure 10.

With increasing CMC content from 0 to 50 wt% the tensile strength and Young’s modulus increase was observed: from ca. 0.2 to 3.4 MPa, and from ca. 0.6 to 4.9 MPa, whereas the elongation at break decreased from ca. 76 to 29%, respectively. However, with higher CMC addition the values of both: tensile strength and Young’s modulus decreased to ca. 2.3 and 2.7 MPa for CMC-based system, respectively, whereas the elongation at break increased up to 50%. As the mechanical properties of the film are determined by the affinity among its components [15], it could be concluded that the molecular structure of the system with equal CMS/CMC ratio is more compact [21], which is probably the result of molecular arrangement and simultaneously crosslinking efficiency as well as hydrogen bonds formation (Figure 2 and Figure 3) [36]. It is worth to mention that the CMC-based films exhibited notably better mechanical properties than CMS-based ones. That is probably the result of higher molecular weight of cellulose than starch derivative [17].

For cassava starch/CMC films (without chemical crosslinking agent addition) other phenomenon was reported, i.e., tensile strength increased gradually with CMC concentration from 0 to 40% [21]. Similar observation was noted for corn starch-based films with CMC content (up to 20%) in the presence of low [34] or higher [2] citric acid load. Tavares et al. [20] tested the mechanical properties of corn or cassava starch/CMC 50/50 wt% films and compared them to the neat starch-based films. Regardless the botanic origin of starch the CMC addition notably improved the tensile strength resistance. On the other side, significant decrease of tensile strength was reported for corn starch/CMC (from papaya peel) systems—i.e., from ca. 31 MPa to ca. 14 MPa—for the film without and with 25% CMC [31].

For sorghum starch/CMC film (in the presence of CA) the highest tensile strength has been noted for 10% CMC (with higher carboxymethylated derivative content the value of this parameter decreased) what has been explained by the strong affinity between polysaccharide molecules [15].

For potato starch/CMS film, the highest tensile strength and Young’s modulus have been determined for the system containing 10 wt% CMS [11], suggesting the best compatibility between components as well as formation of interpenetrated polymer network.

It is important to add, that for most references mentioned above, the degree of substitution of CMC applied was not given. DS is an important factor determining the properties of modified polysaccharide, and this could be the reason why such different effects of CMC addition on the mechanical properties of polysaccharide films were reported.

## 4. Conclusions

The hydrophilic films using carboxymethylated starch and cellulose were obtained by the cast method. The FTIR spectra revealed that crosslinking via ester bridges with citric acid occurred. The effect of CMS/CMC ratio on the physicochemical properties of prepared films was evaluated. Surprisingly, the properties of film with equal CMS and CMC wt% content distinguished from the other systems. For CMS/CMC 50/50 the lowest value of the moisture absorption was noted (18.5% and 60%, respectively) although for CMS/CMC systems: 100/0 and 0/100 the values of this parameter was higher (20% and 23.5%, respectively). Additionally, improved thermal as well as mechanical properties were noted, i.e., the highest tensile strength as well as Young’s modulus: ca. 3.4 MPa and ca. 4.9 MPa, respectively. All above could indicate that the physicochemical properties of CMS/CMC 50/50 system are the result of the molecular structure as well as the interactions between the polysaccharide derivatives macromolecules.

Bearing all above in mind the application of CMS/CMC 50/50 film in a layered system, i.e., in the medical patches, where the hydrophilic CMS/CMC film would be the inner layer (preventing the e.g., burned wound from drying) transferred to an outer layer—exhibiting good mechanical and barrier performance—is planned.

## Figures and Tables

**Figure 1 polymers-12-02447-f001:**
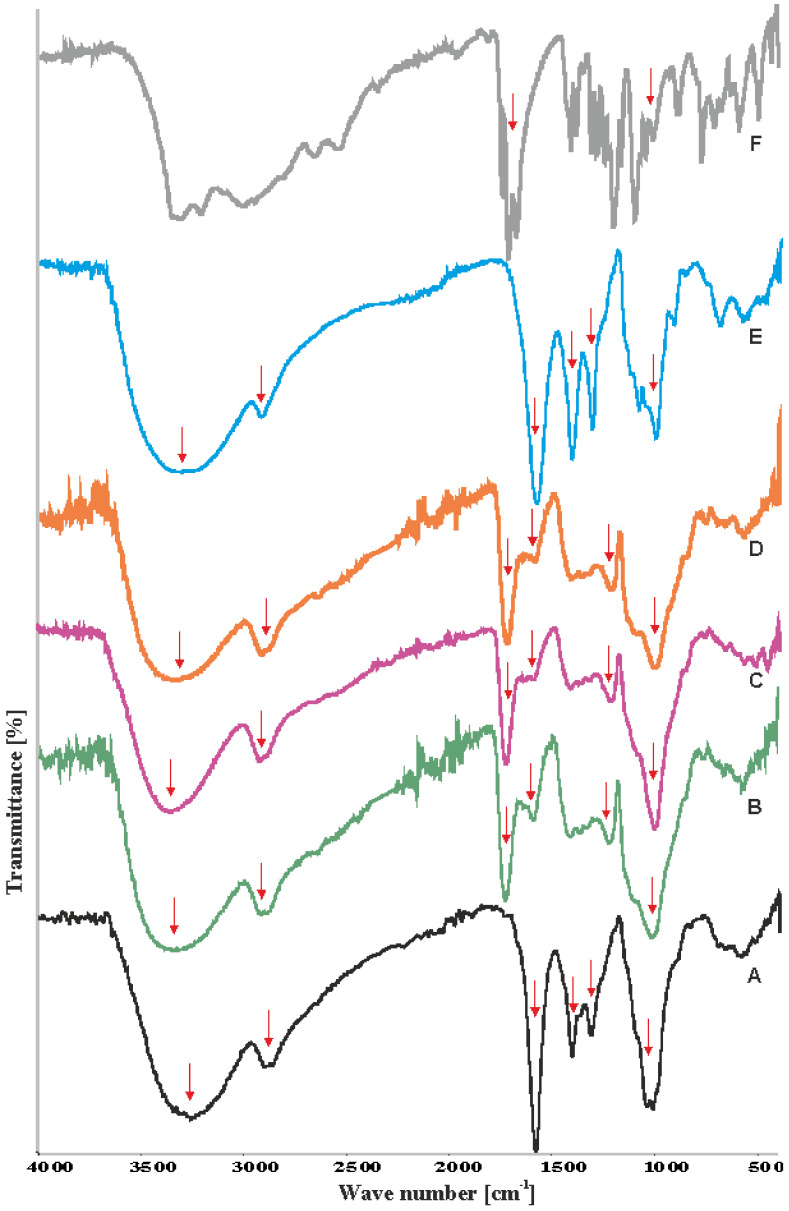
Fourier transform infrared (FTIR) spectra of neat carboxymethyl cellulose (CMC) (curve A), CMS/CMC based films for wt% ratio: 60/40 (B), 50/50 (C); and 40/60 (D), neat carboxymethyl starch (CMS) (E), and citric acid (F).

**Figure 2 polymers-12-02447-f002:**
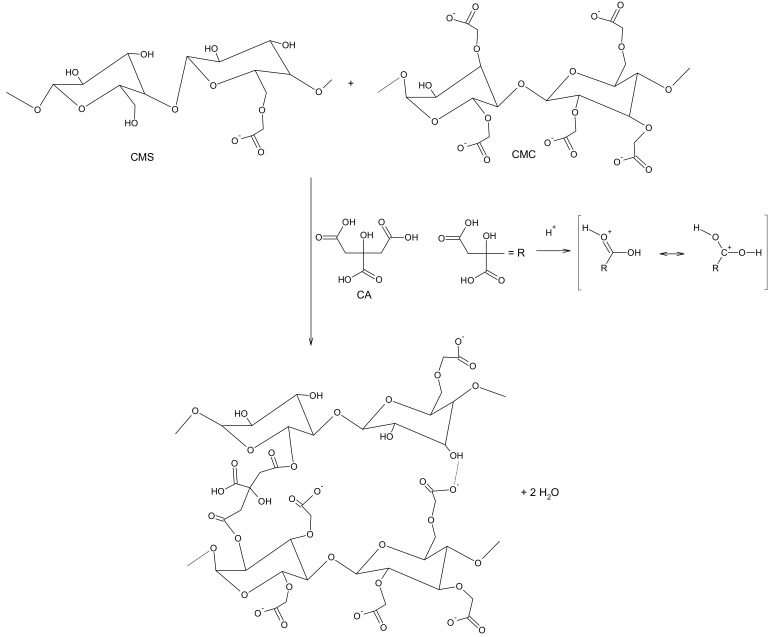
Scheme of CMS/CMC crosslinking reaction with citric acid (CA), for the sake of clarity the reaction for molar ratio equal 1.0 was presented.

**Figure 3 polymers-12-02447-f003:**
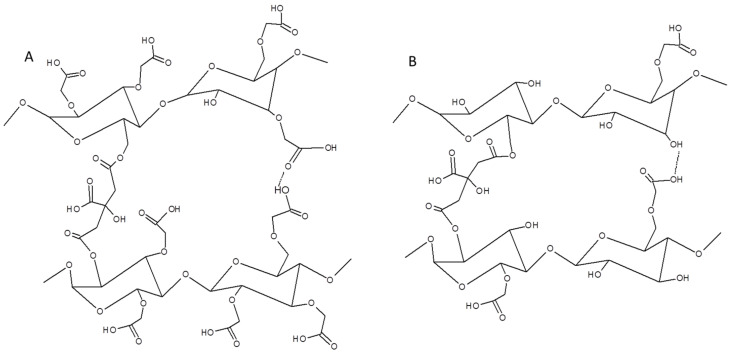
Scheme of interaction between: (**A**) CMC-CMC and (**B**) CMS-CMS molecules.

**Figure 4 polymers-12-02447-f004:**
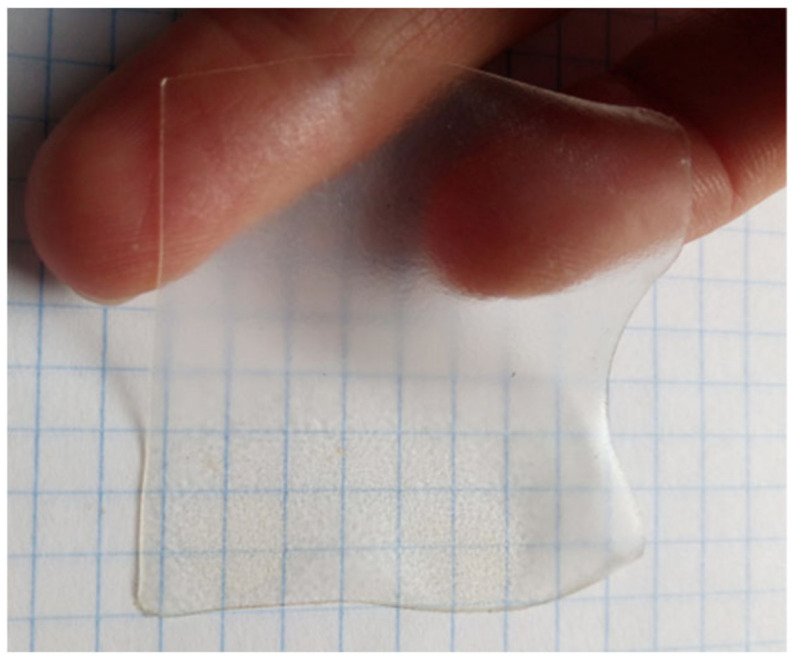
Picture of CMS/CMC 50/50 film.

**Figure 5 polymers-12-02447-f005:**
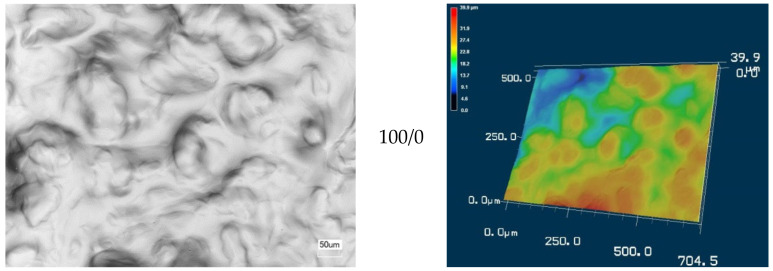
Laser scanning microscopy: (**a**) topographical images and (**b**) 3D images of CMS/CMC-based films.

**Figure 6 polymers-12-02447-f006:**
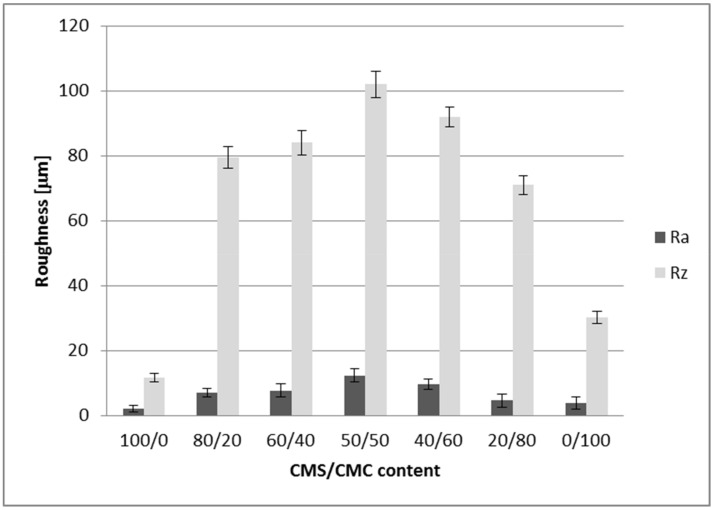
Roughness parameters *R*_z_ and *R*_a_ determined for CMS/CMC-based films.

**Figure 7 polymers-12-02447-f007:**
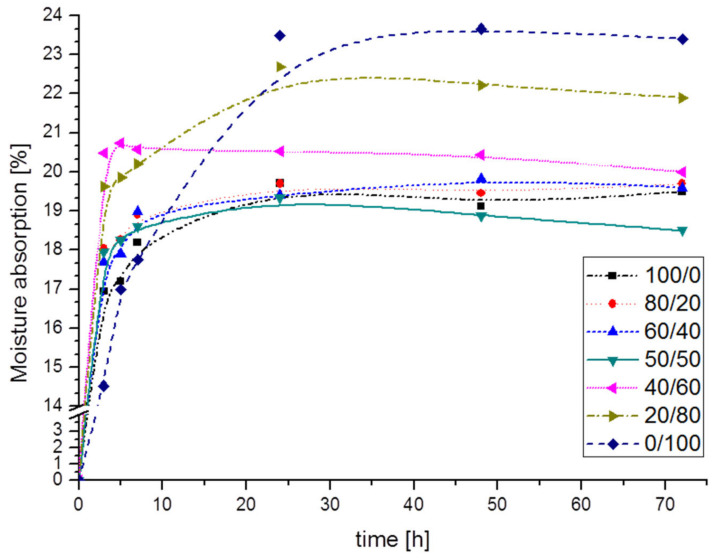
Moisture absorption of CMS/CMC films.

**Figure 8 polymers-12-02447-f008:**
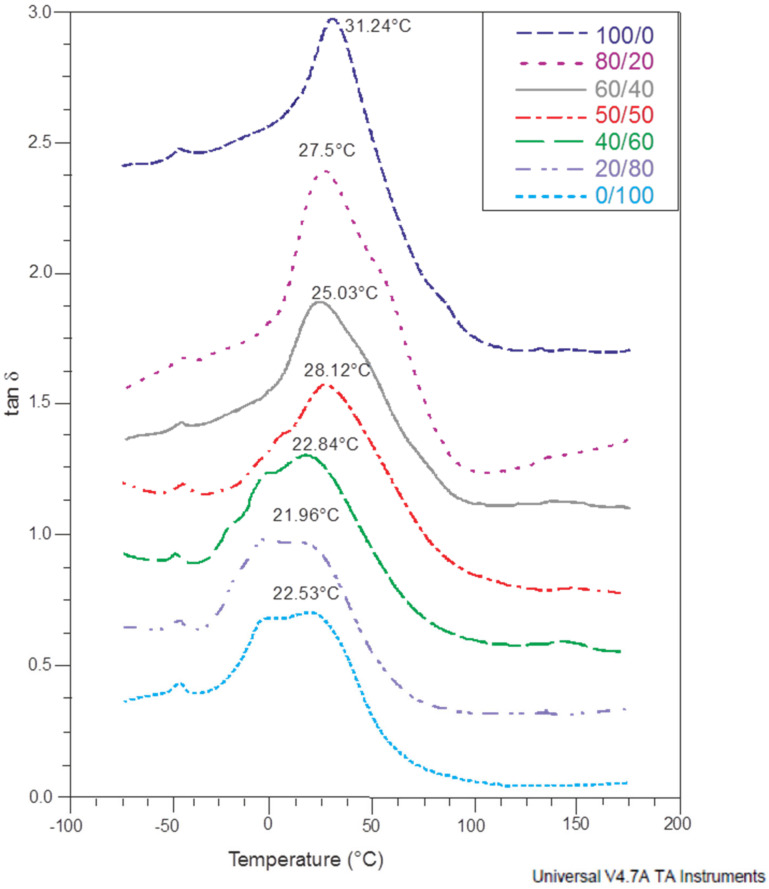
The dynamic mechanical thermal analyses (DMTA) curves of CMS/CMC films.

**Figure 9 polymers-12-02447-f009:**
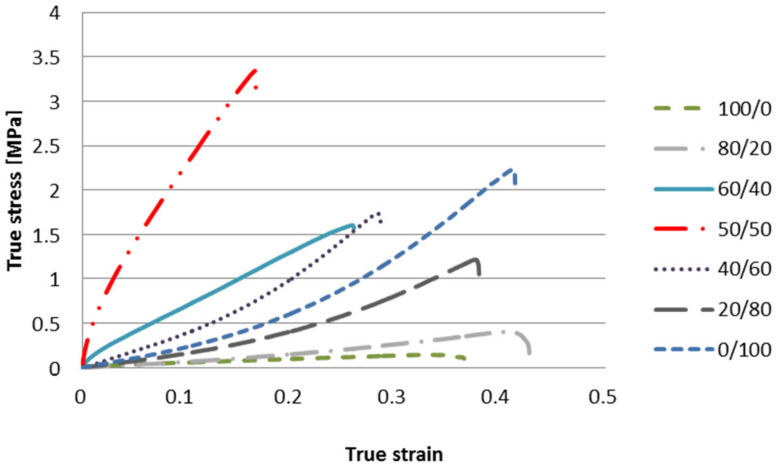
Typical true stress versus true strain curves of CMS/CMC films.

**Figure 10 polymers-12-02447-f010:**
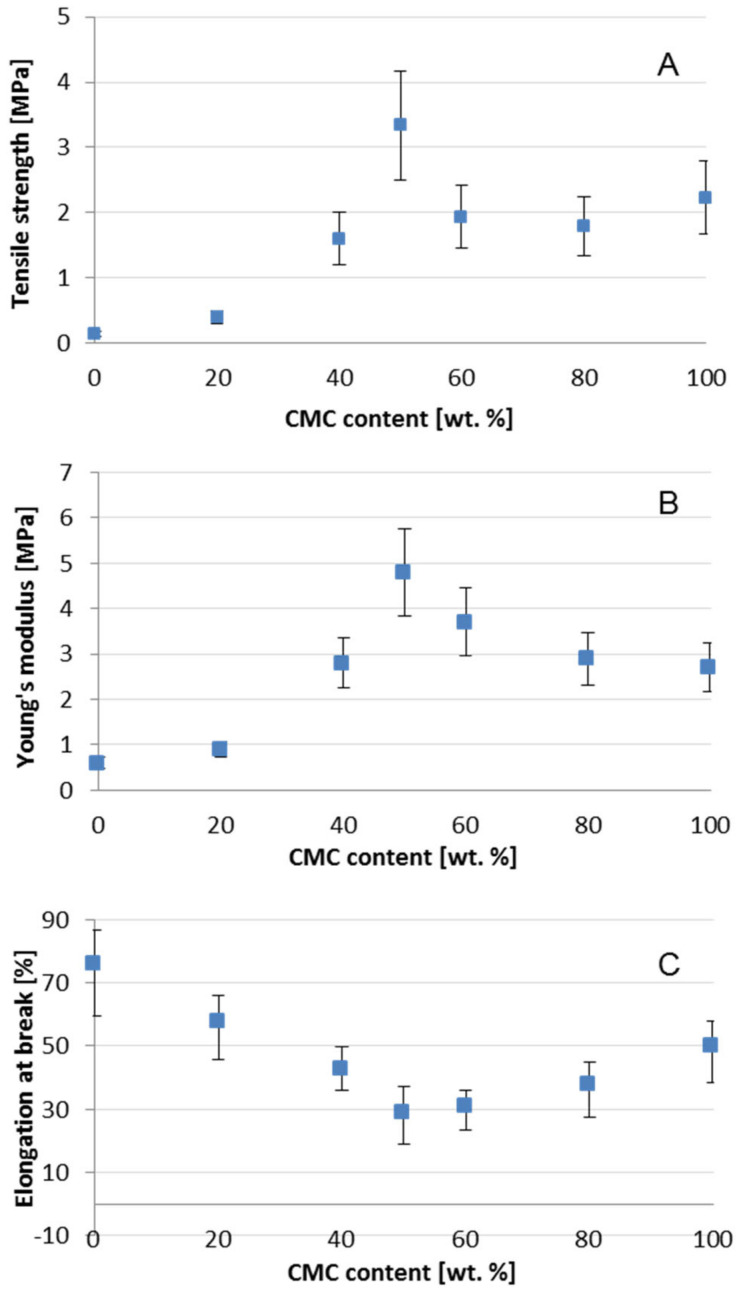
Tensile strength (**A**), Young’s modulus (**B**) and elongation at break (**C**) of CMS/CMC films.

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
