# Peer review of "Hydrophilic Films Based on Carboxymethylated Derivatives of Starch and Cellulose"

_polymers, 2020, doi:10.3390/polym12112447_

Round 1
Reviewer 1 Report
The manuscript (polymers-942815) reports the preparation and characterization of carboxymethylated starch (CMS)/carboxymethylated cellulose (CMC)films at different weight ratios. Critic acid is used to crosslink CMS and CMC through esterification. The moisture absorption, surface roughness, mechanical and thermal properties are studied and the authors demonstrate that the film containing 50/50 CMS/CMC shows the optimal overall performance. Some important issues should be revised before suitable publication in Polymers. Please see below:
- The authors used the critic acid to crosslink CMS and CMC. Comparison study on mechanical properties of CMS/CMC and critic acid-crosslinked CMS/CMC should be conducted.
- What is the concentration of critic acid in the CMS/CMC films? Is it the optimal concentration?
- Page 4. Please label A, B, C, D,E, and F in the curves of Figure 1.
- Page 5, Section 3.2 and Figure 3. How can we distinguish the CMS phase and CMC phase from thoese micrographs? In addition to surface roughness, how about their dispersion and compatibility?
- Some related literature (e.g., Polymer 2016, 107, 200-210; ACS Sustainable Chem. Eng. 2016, 4, 8, 4385–4395) are missing from the references.
Author Response
(polymers-942815)
Reviewer 1
We would like to thank to the Reviewer and truly appreciate your comments, and corrections. Please find the detailed changes and comments below.
The manuscript reports the preparation and characterization of carboxymethylated starch (CMS)/carboxymethylated cellulose (CMC)films at different weight ratios. Critic acid is used to crosslink CMS and CMC through esterification. The moisture absorption, surface roughness, mechanical and thermal properties are studied and the authors demonstrate that the film containing 50/50 CMS/CMC shows the optimal overall performance. Some important issues should be revised before suitable publication in Polymers. Please see below:
- The authors used the critic acid to crosslink CMS and CMC. Comparison study on mechanical properties of CMS/CMC and critic acid-crosslinked CMS/CMC should be conducted.
We obtained the CMS/CMC films without citric acid crosslinking, however those films exhibited very high brittleness and so poor mechanical properties that the mechanical tests could not be performed. This is why could not present the comparison between crosslinked and not-crosslinked films.
- What is the concentration of critic acid in the CMS/CMC films? Is it the optimal concentration?
The citric acid addition was 2 g for each system, which is 30 wt. % on a basis of polysaccharide derivatives mass. It was selected on the basis of our previous studies on CMS-based films crosslinked with CA (J. Polym. Environ. 2019, 27, 1379).
- Page 4. Please label A, B, C, D,E, and F in the curves of Figure 1.
The FTIR curves were labeled, and the characteristic bands of absorption marked.
- Page 5, Section 3.2 and Figure 3. How can we distinguish the CMS phase and CMC phase from thoese micrographs? In addition to surface roughness, how about their dispersion and compatibility?
The CMS and CMC phases could not be distinguished in the micrographs. They exhibit chemically affinity, thus they are miscible and compatible, so in result the molecules interpenetrate each other. The surface roughness is the result of crosslinking: the areas with diester bonds between CMS and CMC occur, however the links between CMS-CMS or CMC-CMC are also possible, especially, when one polysaccharide derivative is in excess. Such uneven crosslinking is the result of roughness variety.
- Some related literature (e.g., Polymer 2016, 107, 200-210; ACS Sustainable Chem. Eng. 2016, 4, 8, 4385–4395) are missing from the references.
The position referring to CMC films with cationic cellulose nanocrystals was included in the manuscript and listed in reference.
Reviewer 2 Report
The authors prepared some carboxymethyl starch/carboxymethyl cellulose crosslinked films with varied physiochemical properties. This work is obviously extended from their previous study (Carbohydrate Polymers 222, 2019, 115014, Carboxymethylated starch and cellulose derivatives-based film as human skin equivalent for adhesive properties testing). All necessary characterizations are provided and the obtained results are relatively reasonable. However, the object of this study is not clear, which makes the manuscript only offers limited scientific and practical significance. Therefore, I recommend it to be accepted for publication in Polymers after major issues are addressed.
- Apparently, CMS/CMC 50/50 film outperforms other systems should be chemically attributed to the crosslinking between CMS and CMC. However, there is no good evidence or reference provided yet to support this mechanism as crosslinking may also happen between CMS molecules or CMC molecules. In addition, since 50/50 is only a weight ratio but not a molar ratio, the proposed scheme is not self-evident based on current data. Please improve the discussion section related to this issue.
- The object of this study is not clear. If the authors were aiming to acquire good mechanical properties of CMS/CMC film, to study the effect of different ratios is insufficient. Deeper understanding of this can be found if the unreacted some carboxylic groups will be crosslinked or quantified (through zeta-potential measurement or conductometric titration), or CMS (CMC) with different carboxymethyl contents or DS values, to support the speculation that “this special ratio CMS/CMC 50/50 the CMS and CMC chains were arranged in the way that favors the crosslinking reaction”.
- The current title is not so accurate and summary that only two carboxymethylated polysaccharides were studied in this work.
- Some figures need to be improved. Figure 1: there is no clear legend of each line to tell curves (A)-(F), and there are only five curves included but with six captions. Figure 3: there are some overlaps in the upper three 3D images and it is also advised to add the camera images of CMS/CMC films. Figure 6: please avoid overlap and use ℃ of the consistent font. Figure 7: please replace all comma with dot.
- The discussion section needs to be improved. For example: it is advised to add related nuclear magnetic resonance data or reference to support the formation of intermolecular diesters shown in Fig.2; CMS/CMC 50/50 (88.3 μm), 40/60 (82.9 μm), 20/80 (73.1 μm) films exhibited lower roughness values than the neat CMC system (90.6 μm); to add a control group of CMS/CMC 50/50 film formed by physical blending may offer extra understanding why this ratio is so special, otherwise it is possible that the moisture absorption of the CMS/CMC 50/50 film is incorrect as other films exhibited growing moisture absorption as increasing the CMC content; more details are needed to form the point as “These results are in correlation with the results of moisture absorption”; the comparisons among CMS/CMC films and different-sourced- starch/CMC films are inexplicable and meaningless if without a summation.
Author Response
(polymers-942815)
Reviewer 2
We would like to thank you for your involvement and contribution in manuscript improvement, and truly appreciate your comments, and corrections. Please find the detailed changes and comments below.
The authors prepared some carboxymethyl starch/carboxymethyl cellulose crosslinked films with varied physiochemical properties. This work is obviously extended from their previous study (Carbohydrate Polymers 222, 2019, 115014, Carboxymethylated starch and cellulose derivatives-based film as human skin equivalent for adhesive properties testing). All necessary characterizations are provided and the obtained results are relatively reasonable. However, the object of this study is not clear, which makes the manuscript only offers limited scientific and practical significance. Therefore, I recommend it to be accepted for publication in Polymers after major issues are addressed.
- Apparently, CMS/CMC 50/50 film outperforms other systems should be chemically attributed to the crosslinking between CMS and CMC. However, there is no good evidence or reference provided yet to support this mechanism as crosslinking may also happen between CMS molecules or CMC molecules. In addition, since 50/50 is only a weight ratio but not a molar ratio, the proposed scheme is not self-evident based on current data. Please improve the discussion section related to this issue.
Thank you very much for pointing this out. Of course, the reaction between CMS-CMS and CMC-CMC molecules occurs, as evidenced for CMS or CMC 100% systems. The crosslinking between CMS-CMS or CMC-CMC molecules would be expected to occur for the CMS/CMC systems when one derivative content was in majority.
The reaction scheme caption was corrected. Moreover, the scheme of interactions between neat CMS as well as neat CMC molecules was added.
Crosslinking of potato starch using citric acid was thoroughly developed within last years, it was evidenced that diester were formed (e.g. LWT-Food Science and Technology, 2016, 69, 334). Crosslinking of CMS molecules wit CA is relatively a new topic, however, basing on the starch citrates formation it could be expected that the mechanism would be very similar, with the except that the crosslinking density would be the factor favoring rather ester, not diester formation.
The CMS/CMC molar ratios for prepared films were added in the “Preparation of CMS/CMC films” section.
The paragraph on solubility in water test was added to discussion:
“It is worth to mentioned that the water solubility tests of prepared CMS/CMC films has been performed. The 1.5 x 1.5 cm2 film samples were placed in 50 ml distilled water, at room temperature. After 48 h all the samples maintained their integrity, i.e., they did not dissolve or break apart during the tests. This indirectly indicates successful crosslinking taking account that both CMS and CMC are soluble in cold water.”
- The object of this study is not clear. If the authors were aiming to acquire good mechanical properties of CMS/CMC film, to study the effect of different ratios is insufficient. Deeper understanding of this can be found if the unreacted some carboxylic groups will be crosslinked or quantified (through zeta-potential measurement or conductometric titration), or CMS (CMC) with different carboxymethyl contents or DS values, to support the speculation that “this special ratio CMS/CMC 50/50 the CMS and CMC chains were arranged in the way that favors the crosslinking reaction”.
The object of this study was to evaluate the effect of CMS/CMC ratio on the physicochemical properties of obtained films. In previous works we tried to obtained the CMS-based films using CMS with various DS. It was reviled, that using CMS with DS values below 0.6 resulted in films with so high brittleness that the mechanical properties were not possible to test. This is why we used only one CMS derivative, with high DS. Moreover, as we expected that the mechanical properties of CMS-based systems were not on the level of other polysaccharides, e.g. starch, we considered potential application of prepared films in rather layered packaging system (where the hydrophilic CMS/CMC film would be responsible for e.g. moisture level of packed product) or medical patches (here the hydrophilic inner layer would prevent the e.g. burned wound from drying) where the outer layer is the one responsible for mechanical performance.
In our opinion the evaluation of unreacted carboxylic group content would not give the information on the crosslinking density as: i) CMS and CMC exhibit different degrees of substitution, ii) citric acid contains three carboxylic groups, it is very unlikely that all of them would react, however it is possible that not all acid molecules act as polysaccharide links (then monoesters instead of diesters can be formed).
- The current title is not so accurate and summary that only two carboxymethylated polysaccharides were studied in this work.
The title was corrected: “Hydrophilic films based on carboxymethylated derivatives of starch and cellulose”.
- Some figures need to be improved. Figure 1: there is no clear legend of each line to tell curves (A)-(F), and there are only five curves included but with six captions. Figure 3: there are some overlaps in the upper three 3D images and it is also advised to add the camera images of CMS/CMC films. Figure 6: please avoid overlap and use ℃ of the consistent font. Figure 7: please replace all comma with dot.
Fig. 1 was corrected: the FTIR curves were labeled, and the characteristic bands of absorption marked.
Fig. 3 (now is Fig. 5): as we wanted to present the 3D views using the same tilt angle the overlapping were performed automatically by the microscope analyzing software. Unfortunately, by changing the tilt the overlaps removed form one picture appeared in another one. We hope, this inconvenience would not affect greatly the clarity of the figure, as the value in 3D informs only on the overall sample topography, not roughness value.
Fig. 6 and 7 (now are fig. 8 and 9) were corrected according to Reviewer’s suggestion.
Additionally, the camera picture of CMS/CMC 50/50 film was added (Fig. 4).
The discussion section needs to be improved. For example: it is advised to add related nuclear magnetic resonance data or reference to support the formation of intermolecular diesters shown in Fig.2; CMS/CMC 50/50 (88.3 μm), 40/60 (82.9 μm), 20/80 (73.1 μm) films exhibited lower roughness values than the neat CMC system (90.6 μm); to add a control group of CMS/CMC 50/50 film formed by physical blending may offer extra understanding why this ratio is so special, otherwise it is possible that the moisture absorption of the CMS/CMC 50/50 film is incorrect as other films exhibited growing moisture absorption as increasing the CMC content; more details are needed to form the point as “These results are in correlation with the results of moisture absorption”; the comparisons among CMS/CMC films and different-sourced- starch/CMC films are inexplicable and meaningless if without a summation.
The reference on citric acid crosslinked starch was supplemented, where the mechanism of diester formation had been presented. Unfortunately, performing NMR measurements is strongly limited at the university because of pandemic situation, and would take much more than 10 days required for submission of reviewed version of manuscript.
The CMS/CMC 50/50 blend (without crosslinking agent addition) was obtained. The moisture absorption of such a film significantly increased within the first hours, and after 24 h was 56% This comparison directly indicates the crosslinking effect in polysaccharide films. The discussion on that topic was developed.
The films roughness (Ra) of CMS/CMC films was presented in Fig. 6, and was as followed: for 50/50 102 mm, for 40/60 92 mm, for 20/80 71 mm, and for the neat CMC 30 mm, thus the roughness of CMC-based system was the lowest one. The x axis values on 3D view referred to the general film topography. For roughness evaluation the line profiles were performed, and the sum of the height of the highest profile peak and the depth of the deepest profile valley within an individual measuring distance was determined (Rz); the Ra were the arithmetically determined values.
The sentence: “These results are in correlation with the results of moisture absorption” was removed from the manuscript.
A short summary on the CMC presence effect on mechanical properties of polysaccharide films in section 3.5. was added.
Reviewer 3 Report
The present manuscript involves the development of films based on potato starch (CMS) and cellulose (CMC). Before film obtention, the polysaccharides were crosslinked with citric acid under reactions of etherification. The effect of CMS/CMC ratios on physicochemical properties of the films was evaluated.
The authors presented interesting research to be published in the Polymers Journal.
However, I have some recommendations to be performed to increase the quality of the final manuscript.
--> It is necessary to assign each FTIR spectra for CMC, and the different CMS/CMC based films in its respective letters.
-->It would be interesting if the authors could perform an illustrative scheme presenting the interactions between CMS and CMC, to elucidate more the explanation of section 3.5.
The main problem with this manuscript is the lack of application. The authors discussed the possibility of application in agriculture or pharmacy. However. I think that is necessary to perform an application in order to enrich the results. Only the physicochemical characterization is a poor result considering the potentiality of this kind of systems.
Author Response
(polymers-942815)
Reviewer 3
We would like to thank to the Reviewer and truly appreciate your comments, and corrections. Please find the detailed changes and comments below.
The present manuscript involves the development of films based on potato starch (CMS) and cellulose (CMC). Before film obtention, the polysaccharides were crosslinked with citric acid under reactions of etherification. The effect of CMS/CMC ratios on physicochemical properties of the films was evaluated.
The authors presented interesting research to be published in the Polymers Journal.
However, I have some recommendations to be performed to increase the quality of the final manuscript.
--> It is necessary to assign each FTIR spectra for CMC, and the different CMS/CMC based films in its respective letters.
Fig. 1 was corrected: the FTIR curves were labeled, and the characteristic bands of absorption marked.
-->It would be interesting if the authors could perform an illustrative scheme presenting the interactions between CMS and CMC, to elucidate more the explanation of section 3.5.
The scheme considering interactions between CMS-CMS and CMC-CMC molecules was added (Fig. 3), according to Reviewer’s suggestion.
The main problem with this manuscript is the lack of application. The authors discussed the possibility of application in agriculture or pharmacy. However. I think that is necessary to perform an application in order to enrich the results. Only the physicochemical characterization is a poor result considering the potentiality of this kind of systems.
Thank you for pointing this aspect. In this manuscript we focused on the effect of CMS to CMC ratio on the physicochemical properties of obtained films. It is also the screening test allowing to select the system exhibiting the best performance for the application tests. We consider potential application of prepared films in a layered system, first in the medical patches (here the hydrophilic CMS/CMC film would be the inner layer preventing the e.g. burned wound from drying), thus the mechanical properties is not the essential parameter, as the outer layer is the one that should exhibit good mechanical performance. Moreover, we plan to test such a system according to transdermal permeation – by introducing into CMS/CMC layer the active drug, like analgesic agent. The scope of results of mentioned research would exceed the Journal requirements, thus we consider other paper on that subject.
The sentence supplementing the planned application in “Conclusion” section was added.
Round 2
Reviewer 1 Report
The authors have adequately addressed my concerns. I recommend it for publication in Polymers.
Reviewer 2 Report
The authors have replied the comments in detail. This work can be accepted now.